# Assessing affective valence and activation in resistance training with the feeling scale and the felt arousal scale: A systematic review

**Vasco Bastos**[1,2], **Filipe Rodrigues**[3,4], **Paul Davis**[5]*, **Diogo Santos Teixeira**[1,2]

**1** Faculty of Physical Education and Sport (ULHT), Lusófona University, Lisbon, Portugal, **2** Research Center in Sport, Physical Education, and Exercise and Health (CIDEFES), Lisbon, Portugal, **3** ESECS—Polytechnic of Leiria, Leiria, Portugal, **4** Quality of Life Research Center (CIEQV), Santarém, Portugal, **5** Department of Psychology, Umeå University, Umeå, Sweden

* paul.davis@umu.se

**Data Availability Statement:** All relevant data are within the paper and its Supporting Information files.

## Abstract

Evidence suggests affective responses to exercise can influence exercise adherence. However, there is a limited understanding of how and when to measure core affect in resistance training. As such, the objective of this systematic review was to analyze how the Feeling Scale and/or the Felt Arousal Scale have been used in resistance training to assess core affect. Focus was given to the contextual feasibility, timing, and frequency of assessment. A search in PubMed, SPORTDiscus, and PsycINFO databases was conducted (last search date July, 2022) with the purpose of including experimental and non-experimental studies, utilizing the Feeling Scale and/or the Felt Arousal Scale in resistance training, and focused on apparently healthy individuals of any age. Twenty-seven studies (N = 718 participants) published between 2009–2022 were qualitatively analyzed. Both scales appeared to be able to detect core affect within a wide array of intensities, ages, and equipment. As for the timing and frequency of measurement, no apparent standardization was evident. The use of the Feeling Scale, the Felt Arousal Scale, or both, to measure core affect appears to be feasible in resistance training practices. However, a lack of methodological background raises concerns regarding the quality of previous studies' assessments and comparisons of results across studies.

## Introduction

Resistance Training (RT) is a mode of exercise that promotes a plethora of health-related benefits [1, 2]; it can be particularly impactful in osteoporosis and sarcopenia prevention, as well as muscle mass maintenance [3, 4]. Considering these benefits, further research regarding the positive impact of RT on health parameters is relevant for public health interventions [5, 6]. However, a major issue for individuals in the undertaking of physical activity (PA), and thus RT, is the lack of motivation to perform regular exercise. A concerning portion of the world's population does not meet PA recommendations (87% of adolescents and 27.5% of adults [7]), and poor motivation ranks as one of the primary reasons for opting out of physical activity [8].

**Funding:** The author(s) received no specific funding for this work.

**Competing interests:** The authors have declared that no competing interests exist.

Thus, implementing effective behavior change techniques to help individuals become motivated to engage in RT, and exercise in general, is of paramount importance. To this end, the study of affect in PA may be a relevant line of research to pursue [9].

## The role of affect in exercise adherence

Grounded in hedonic principles, 'affectivism' (i.e., the study of affect) is a line of thought postulating that people tend to engage in activities they consider pleasurable while avoiding those wherein they experience pain and displeasure [9, 10]. Accordingly, several studies have shown that affect and affective dependent variables (e.g., anticipated affective response to exercise; implicit attitude) demonstrate some predictive value to exercise adherence-related variables (e.g., habit, intention, frequency), and could advance current theoretical models of exercise behavior [11–14].

As an example, in the Affect and Health Behavior Framework (AHBF [15]) the affective response to exercise is highlighted as a determinant in the sustainability of such behavior. According to this framework, core affect (i.e., an elementary, non-reflective feeling, consciously available to the individual) is the central aspect of the affective response to exercise and has been suggested to be a reliable predictor of exercise adherence. To maintain ecological validity, the affective response is best measured during or immediately after the activity, since it can only be experienced *in vivo* [15, 16]. However, this assessment can be difficult to perform given the dynamic nature of particular forms of PA (e.g., assessing core affect during the execution of the bench press exercise). Notwithstanding, some advancement in the assessment of exercise-related affect has been made, but several methodological clarifications for proper use and interpretation are needed [17].

The affective response dynamics of a given exercise session can be evaluated through the measurement of two core dimensions of affect: affective valence (perceived pleasure-displeasure) and arousal (perceived activation) [16, 18]. For the purpose of measuring these constructs, the Feeling Scale (FS [19]) and the Felt Arousal Scale (FAS [20]) were developed and have been used extensively in exercise settings [17, 21]. Both scales can be used in conjunction and interpreted in relation to the circumplex model of affect [22] to map affective fluctuations throughout an exercise session [16, 18]. Information derived from the FS and FAS can facilitate adjustments during exercise sessions to promote a more pleasurable experience, and potentially contribute to sustainable exercise programs [23, 24].

## Measuring core affect in resistance training

Exercise guidelines first recommended the measurement of affective responses (e.g., with the FS) to exercise over a decade ago [25]. Currently, the latest edition of the ACSM's guidelines for exercise prescription and associated recommendations are still present in the form of *'affect regulation'* (p. 455 [1]). However, there is persistent uncertainty regarding how to effectively *assess* and *regulate* affect in exercise prescription (i.e., operationally; how and when to evaluate, interpret, and make the necessary adjustments). This knowledge gap is particularly prevalent within RT research literature, and has been noted in recent works [17, 26].

One important limitation in RT studies concerns the timing of affect measurement (i.e., when [27]). In their review, Evmenenko and Teixeira [17] report that no apparent standardization of measurement timing exists in the RT research that plots core affect within the circumplex model of affect. In particular, the timing of assessment is of notable concern due to a possible 'affective rebound' phenomenon (i.e., improvement of the affective response after exercise termination) that has been well documented in aerobic activities [16], as well as emerging evidence from the limited studies on RT [28–30]. Inappropriate timing of

assessment may lead to the measurement of confounding variables beyond core affect and promote the reporting of biased results. Thus, one critical primary issue to address is to clarify *when* to assess core affect during RT.

Additionally, the frequency of FS/FAS assessments required for an adequate interpretation of affective response to exercise is another concern that remains unaddressed within RT research [17]. As suggested by Haile et al. [31] as well as Zenko and Ladwig [32], measurements should be recorded at regular intervals to effectively assess the exerciser's affective response, whilst simultaneously avoiding the burden of excessive assessment. It is proposed that this balance is influenced by several variables (e.g., exercise experience, type, health status), making the frequency of measurement an important aspect in the contextual application of these scales. However, no clear indication in the research literature or exercise guidelines has emerged regarding the frequency of assessment in RT; thus, the intention of promoting positive affect in this form of exercise is impeded. Given that the advancement on this frequency of application issue is heavily dependent on methodological clarifications regarding *when* to assess core affect, assessment standardization efforts must come forward to enhance further research endeavors.

It also warrants consideration that contextual factors such as exercise intensity and exercise mode (e.g., machines, free-weights, calisthenics), as well as individual factors (e.g., age, sex, exercise experience), may also be relevant for advancing understanding of core affect assessments in RT. This line of study would attend to the concerns raised by previous reports [28, 32, 33] that have highlighted the potential interaction of contextual and individual variables influencing affective responses, as well as evaluate the feasibility of using these scales across various conditions.

## Present study

Given the need to reduce inconsistency and promote specific guidelines for affect assessment during RT activities, more research to improve methodological quality is warranted [17, 28, 29]. As such, the main objective of the present review was to analyze how the FS and/or the FAS have been used in RT to assess core affect. More specifically, we aimed to analyze how the FS and/or the FAS have been used in RT practices, and particularly, how the timing and frequency of assessment have been made, with the purpose of understanding these scale's value and contextual feasibility for core affect measurement in RT. To this end, their feasibility, focused on the contextual and specific characteristics of this exercise type (intensity, expression, apparatus) and sociodemographic variables (age, sex, experience), in conjunction with the timing and frequency of assessment of both scales, will be explored.

## Method

This review was undertaken following the recommendations proposed by the PRISMA protocol [34] and is registered in PROSPERO with the number CRD42022332897.

### Eligibility criteria

The present review applied the following inclusion criteria: (1) experimental, longitudinal, and cross-sectional studies; (2) published in a peer-reviewed journal or as gray literature until July 30 of 2022; (3) written in English; (4) utilizing the FS and/or the FAS in RT exercise; and (5) focused on apparently healthy individuals of any age. The exclusion criteria were as follows: (1) populations with mental disease; (2) body mass index $> 34.9$ Kg/m$^2$; (3) mixed exercise programs (i.e., circuit training and similar exercise program structures); (4) instrument validation studies; and (5) review studies.

## Information sources and search strategy

A wide search of the literature was conducted from February 1 of 2022, to July 30 of 2022 on the following databases: PubMed (host: MEDLINE): last search run July 2022; SportDISCUS (host: EBSCO): last search run July 2022; and PsycINFO (host: EBSCO): last search run July 2022. The PICOS strategy was applied, and the search was executed with the following entries in each individual database: (((physical AND (exercise OR activity)) AND (feeling scale OR felt arousal scale) AND ((resistance OR strength) AND training))). A sample of the PubMed search strategy can be found in S1 File.

Bibliographic references from related research and other sources were examined with the purpose of including more studies that potentially met the inclusion criteria (the last search was conducted on July 30 of 2022).

## Selection process

Three independent reviewers (VB, DT, and FR) were involved in the article selection process. All reviewers were trained in the study procedure and disagreements were resolved in group discussion and consensus. At Level I, titles and abstracts of all identified records from the database search were manually screened, analyzed, and checked against eligibility criteria. Full-text publications of every study not eliminated in the previous screening were retrieved for complete review at Level II. At this stage, the authors read the full-text publications to guarantee that the inclusion criteria were met and no exclusion criteria were present. The complete search and screening process is illustrated in Fig 1.

## Data collection process and data items

Two reviewers (VB and DT) independently conducted the data collection process. For general description (Table 1), the following characteristics were extracted from the included studies: (1) bibliographic information (authors, year of publication, country of research); (2) study design; (3) sample size; (4) sample characteristics; (5) intervention; (6) measures; (7) statistical analyzes; and (8) outcomes of interest. A data extraction sheet was made in Excel to summarize all data of interest from the studies. For a summary of the main characteristics of interest (Table 2), the following data were collected: (1) sample size; (2) sex; (3) location; (4) age; (5) effect size and/or power calculation; (6) applied instruments; (7) affective response and arousal measurement prior training with FS/FAS; and (8) timing of measurement.

## Study risk of bias assessment

The risk of bias assessment was conducted with the recommended tools for this purpose by the Cochrane Collaboration (Risk of Bias 2 [35]). This instrument assesses the risk of bias in randomized controlled trials according to the following six domains: randomization process; deviation from intended interventions; missing outcome data; measurement of the outcome; selection of the reported results; and overall bias. Each domain was judged as 'low risk', 'some concern', or 'high risk'. An extension of this tool for crossover trials was also used to assess the included studies with this specific design. The domains assessed are the same as its counterpart for randomized controlled trials, with the addition of the 'bias arising from period and carry-over effects' domain. Lastly, the risk of bias in the quasi-experimental studies was evaluated with the Risk Of Bias In Non-randomized Studies of Interventions (ROBINS-I [36]). With this tool, reviewers give a score of 'low risk'; 'moderate risk'; 'serious risk'; 'critical risk'; or 'no information' to each of the following domains: 'confounding'; 'selection of participants into the study'; 'classification of interventions'; 'deviations from intended intervention'; 'missing data'; 'measurement of outcomes'; and 'selection of the reported outcomes'. Two reviewers

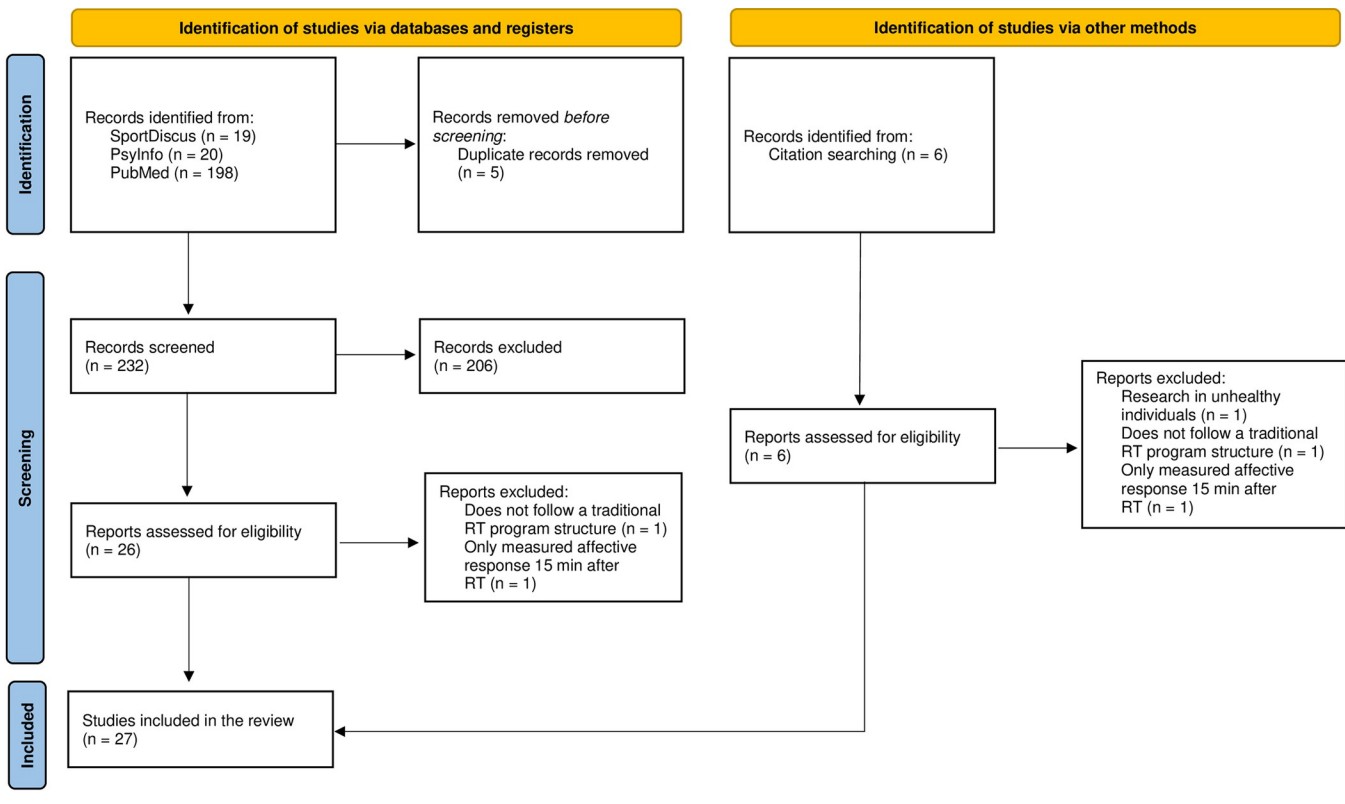

**Fig 1. Study flow-chart.**

(VB and FR) independently analyzed the included studies and disagreements were resolved by consensus. If disagreements persisted, a third reviewer (DT) was consulted to resolve the discrepancies. All reviewers were debriefed and instructed prior to the use of each risk of bias assessment tool. Lastly, the robvis online application [37] was used to create figures that present a more detailed overview of each included study's risk of bias (S2–S4 Files).

## Results

### Study selection

A total of 237 studies were identified during the database search for potential inclusion. After the removal of five duplicate records, 232 studies entered the screening process. Following a meticulous read-through of the title and abstract of every record, 206 were ultimately excluded for reasons described in Fig 1. Of the 26 resulting studies, two additional studies were excluded after full-text reviews. One study was excluded due to not following a traditional RT program [38], and another study was excluded due to only measuring affective response 15 minutes after the RT session [39]. Analysis of the bibliographical references of the selected studies revealed six additional studies that were identified to be potentially relevant. Of these studies, three met the inclusion criteria, and three were excluded [40–42], resulting in a final number of 27 studies, published or accepted for publication until July 31 of 2022, that proceeded to an in-depth analysis.

### Study characteristics

A synthesis of the data collected from the 27 studies that comprise this review can be observed in Tables 1 and 2 (organized in alphabetic order according to the first author's surname). All

**Table 1. Descriptive characteristics of the studies and main outcomes.**

| Author(s) | Location | Design | Size | Mean age and standard-deviation | Intervention | Measures | Analysis | Outcomes | Bias risk |
|---|---|---|---|---|---|---|---|---|---|
| Alves et al. (2014) [43] | Brazil | Quasi-experimental study | n = 11 | 13.7 ± 2.1 | A training session consisting of three exercises: bench press, leg press and barbell curl with 3 sets of 10 repetitions at a self-selected intensity with a 1-minutes interval for rest 1-RM was measured in a previous session | FS after each set | Repeated Measures ANOVA | The three exercises resulted in a positive affective response in this sample of obese adolescent women, with the bench press and leg press presenting a significantly higher result than the barbell curl. | Serious |
| Alves et al. (2017) [44] | Brazil | Quasi-experimental study | n = 14 | 39.2 ± 11.1 | A training session at a self-selected intensity consisting of 3 sets x 10 reps of 5 exercises: bench press, leg extension, front lat pulldown, barbell curl, and leg curl. A preparatory session took place before the training session | FS after each set and 30 minutes after the exercise session | One-way ANOVA; paired t-test | All exercises presented a positive affective response, with the barbell curl resulting in the least pleasurable response and the leg curl presenting a lower value in comparison with the leg extension No significant differences between the in-session and 30' post-session affective response | Serious |
| Andrade et al. (2022) [27] | Portugal | Quasi-experimental study | n = 33 | 36.42 ± 7.72 | The back squat and bench press were performed at 60% RM (2 sets of 15–17 reps), 75% RM (3 sets of 8–10 reps) and 90% RM (4 sets of 5–6 reps) 1-RM was measured in a previous session | FS & FAS at three time points: intra-set, immediately after the set and 5 to 10 seconds afterward PRETIE-Q-PT | Shapiro-Wilk test; Levene's test; Mauchly's test; repeated Measures ANOVA; Bonferroni correction; η2 effect size | No differences among the 3 different affective response assessment time points and %RM | Serious |
| Bastos et al. (2022) [26] | Portugal | Quasi-experimental study | n = 43 | 34.69 ± 6.71 | Six resistance training exercises (pulldown, back squat, bench press, deadlift, dumbbell shoulder press and leg extension) and two bouts of aerobic training (preparatory phase and cool-down) comprised the exercise session A preparatory session took place before the main exercise session | FS & FAS after the third set of each resistance training exercise (after reaching concentric failure) and after the two aerobic bouts PRETIE-Q-PT; PACES; RPE/RIR | Shapiro-Wilk test; Levene's test; Mauchly's test; repeated Measures ANOVA; Bonferroni correction; η2 effect size; Kruskall-Wallis test | The affective response remained in the high pleasure/activation quadrant of the circumplex model of affect for the 6 resistance training exercises. Results support that a single measurement with the FS and the FAS can be enough in assessing affective response in an RT session with experienced exercisers | Serious |

*(Continued)*

**Table 1.** (Continued)

| Author(s) | Location | Design | Size | Mean age and standard-deviation | Intervention | Measures | Analysis | Outcomes | Bias risk |
|---|---|---|---|---|---|---|---|---|---|
| Bellezza et al. (2009) [45] | USA | Quasi-experimental study | $n = 29$ | 20.9 ± 1.9 | Two training sessions with either a large to small (chest press, leg press, rows, leg extension, overhead press, hamstring curl, bicep curl, calve raise and triceps extension) or small to large (the reverse) exercise order 1-RM was measured in a previous session | FS & FAS before, during (mid-point), immediately after and 10 minutes after the training session RPE | Repeated measures GLM; T-test; Bonferroni correction | The small to large muscle groups protocol showed a more pleasurable response during and 10 minutes post-session Both protocols presented an increase in affective valence and activation, with a low-activation pleasure response immediately after exercise | Moderate |
| Carraro et al. (2018) [33] | Italy | Quasi-experimental study | $n = 30$ | 23.8 ± 5.1 | Two training sessions on 2 separate days, one with 3 machines (chest press, shoulder press machine and leg press) and the other with three free weight exercises (bench press, front military press and squat) | FS & FAS were applied immediately after the exercise session RPE; PACES | T-test; Cronbach's α; Pearson's correlations | Free weights resulted in increased pleasantness and activation compared with machine training | Moderate |
| Cavarretta et al. (2019) [28] | USA | Quasi-experimental study | $n = 28$ | Males ($n = 7$): 22.6 ± 4.6 Females ($n = 21$): 23.4 ± 8.6 | Two workouts consisting of 4 machines (leg press, row, chest press and leg curl) or 4 free weight (goblet squat, row, bench press and stiff-leg deadlift) exercises for 3 sets of 9–11 repetitions at 80% 10RM | FS was measured before, during (both intra-set and inter-set), and at 5-min and 30-min post-exercise | Repeated Measures GLM | A more positive affect was verified 5-min and 30-min post exercise, compared to before. Additionally, affect was more positive at 5-min compared to 30-min post ($p = 0.015$) and higher for the inter-set measurement compared to the intra-set measurement | Serious |
| Cavarretta et al. (2022) [46] | USA | Quasi-experimental study | $n = 29$ | Males ($n = 8$): 22.3 ± 4.4 Females ($n = 21$): 23.4 ± 8.6 | A 10RM test was completed for 4 machine exercises (leg press, row, chest press and leg curl) in one session and 4 free-weight exercises (goblet squat, row, bench press and stiff-leg deadlift) in another session | FS was measured after each successful 10RM attempt RPE/RIR | ANOVA; Fisher's LSD pairwise comparison | Affect became less positive only at 100% 10RM compared with all other loads. The affective response was also more positive for upper-body exercises compared to lower-body exercises and more positive for machines compared to free-weights | Moderate |

*(Continued)*

**Table 1.** (Continued)

| Author(s) | Location | Design | Size | Mean age and standard-deviation | Intervention | Measures | Analysis | Outcomes | Bias risk |
|---|---|---|---|---|---|---|---|---|---|
| Chang & Etnier (2009) [47] | USA | Randomized controlled trial | n = 68 | 25.95 ± 3.2 | Four randomly assigned groups: control, 40%, 70%, or 100% 10RM. The intervention groups performed 2 sets of 10 repetitions of 6 exercises: bench press, right and left rowing, lateral arm raises, and right and left arm curl. | FS & FAS were measured at baseline (after sitting quietly in a room for 15 minutes), and immediately after each of the six exercises RPE | ANOVA; MANOVA; Tukey post hoc comparison | Affect did not differ between treatment groups. Activation showed significant differences between groups with a tendency for higher values in the groups with the higher load intensity. | Some concerns |
| Chmelo et al. (2009) [48] | USA | Quasi-experimental study | n = 32 | 21 ± 1.4 | Participants took part in two sessions of eight exercises either with or without mirrors in a randomized fashion. The first seven exercises (chest press, rows, squats, lateral raises, bicep curls, triceps extensions and dead lifts) were completed in two sets (60% and 100% 10RM) of 10 repetitions with the last exercise (crunches) being performed to failure | FS & FAS were measured prior to, during (middle point, after the lateral raises exercise) immediately following and 15 minutes post-exercise AD ACL | Repeated measures GLM; Fisher´s LSD | Affect was more pleasant and activated during and following exercise, but did not differ between the mirrored or no mirrors conditions | Moderate |
| Elsangedy et al. (2016) [49] | Brazil | Quasi-experimental study | n = 12 | 35.8 ± 5.8 | Subjects participated in four sessions: one familiarization session; two for 1-RM tests; and a RT session where they performed 3 sets, 10 repetitions each, with a self-selected load in seven exercises (leg press, chest press, seated rows, knee extensions, overhead press, bicep curl, and triceps pushdown). | FR & OMNI-RES were applied after each set of the self-selected RT session | Descriptive statistics; intra-class correlation coefficient; coefficient of variation | Sedentary male subjects self-selected approximately 55% of 1-RM, a value above the recommendations to increase strength in sedentary individuals, but below the intensity recommended to improve strength in novice to intermediate exercisers. Mean affective response for all exercises remained between 0–1 points, with a large inter-subject variability. | Serious |

(Continued)

**Table 1.** (Continued)

| Author(s) | Location | Design | Size | Mean age and standard-deviation | Intervention | Measures | Analysis | Outcomes | Bias risk |
|---|---|---|---|---|---|---|---|---|---|
| Elsangedy et al. (2018) [50] | Brazil | Quasi-experimental study | $n = 16$ | 39.7 ± 7.5 | Three familiarization sessions, two 1RM sessions and 16 RT sessions (4 for each FS descriptor; randomized) were performed The RT exercises were the leg press, chest press, knee extension, and seated bicep curl (3 sets of 10 repetitions) | Four FS descriptors were utilized to select the intensity of load: "very good" (FS+5), "good" (FS+3), "fairly good" (FS+1), and "fairly bad" (FS-1) OMNI-RES scale | Three-way analysis of variance ANOVA; Greenhouse-Geisser correction; eta squared; Bonferroni correction | The lower the FS descriptor the higher the weight lifted | Moderate |
| Elsangedy et al. (2021) [51] | Brazil | Randomized controlled trial | $n = 32$ | 66.0 ± 3.0 | Experimental group performed a self-selected resistance training program three times per week over 12 weeks The RT exercises performed were bench press, leg press, lateral pulldown, knee extension, lateral shoulder raise, knee curl, biceps curl, and triceps pushdown (3 sets of 15 repetitions) | FS was applied at the end of each set of every exercise during all training sessions OMNI-RES | Generalized linear model; one-way repeated-measures analysis of variance; boxplots; Shapiro-Wilk's test; Mauchly's test; Greenhouse-Geisser epsilon correction; Bonferroni correction | All components of functional capacity improved compared to the control group The exercise sessions were perceived as pleasant and of low to moderate effort | High |
| Emanuel et al. (2021) [29] | Israel | Quasi-experimental study | $n = 20$ | Males ($n = 10$): 28 ± 6 Females ($n = 10$): 32 ± 6 | Three sessions of 3 sets to task failure with either (1) 70% 1-RM bench press, (2) 70% 1-RM squat (squat-70%), or (3) 80% 1-RM squat (squat-80%) 1-RM was measured in a previous session | FS was applied after each and every repetition across all sets RPE; ROF scale | Mixed regression models; mixed analysis of variance; Mauchly's test; Greenhouse-Geisser correction; Holm corrected for multiple comparisons | FS ratings predicted proximity to failure and bar velocity reduction in all three conditions It could be observed that the timing of FS measurement can considerably influence the results | Moderate |

(*Continued*)

**Table 1.** (*Continued*)

| Author(s) | Location | Design | Size | Mean age and standard-deviation | Intervention | Measures | Analysis | Outcomes | Bias risk |
|---|---|---|---|---|---|---|---|---|---|
| Emanuel et al. (2021) [52] | Israel | Quasi-experimental study | *n* = 20 Effect size and power calculations | Males (*n* = 10): 30 ± 4 Females (*n* = 10): 29 ± 4 | Two sets of squats followed by two sets of bench press to task failure, using 70% or 83% of 1-RM, were completed in two sessions 1-RM was measured in a previous session | FS was measured within 10 seconds after set completion RPE; HOF; exercise enjoyment scale; load preference | Linear, and quadratic mixed models; mixed regression models; Mauchly's test; Greenhouse-Geisser correction; Holm corrected for multiple comparisons | RPE scores accurately reflected reaching task failure across loads and conditions. The lack of significant differences in affective valence, rating of fatigue, enjoyment, and load preference between load conditions indicate that when sets are taken to task failure, loads can be selected based on individual preferences | Moderate |
| Ferrreira et al. (2013) [53] | Brazil | Quasi-experimental study | *n* = 14 No effect size or power calculations | 68.5 ± 4.6 | Three sessions of concentric, eccentric, or dynamic training were applied in a randomized order. Each session consisted of 5 exercises (lying supine, leg extension, front pulley, leg curl, and side lifting) performed for 3 sets of 8–10 repetitions 1-RM was measured in a previous session | FS & FAS were presented during the rest interval between sets, in a randomized order OMNI-RES | Repeated measures ANOVA; Bonferroni correction; Greenhouse-Geisser correction; partial eta squared | The affective and RPE responses were similar between the different muscle actions, with the exception of the front pulley exercise in the eccentric training, which exhibits a better perceptual and affective (albeit non-significant) response | Moderate |
| Focht et al. (2015) [54] | USA | Quasi-experimental study | *n* = 20 | 23.15 ± 2.92 | Three sessions involving 3 sets of 10 repetitions of 5 exercises (leg extension, chest press, leg curl, and lat pull-down) using loads of 40% 1-RM, 70% 1-RM and a self-selected load. 1-RM was measured in a previous session | FS was applied before, during (after the third set of each exercise), and after (immediately and 15 minutes postexercise) each session Intention; self-efficacy | Repeated-measures ANOVA; univariate ANOVA; Huynh and Felt test; LSD test; bivariate correlations | Self-selected and imposed load RT resulted in comparable improvements in post exercise affect, when compared to baseline. However, the 70% 1-RM condition showed a decrease in affect from baseline during the exercise session, only improving after the termination of the training session. In contrast, both the self-selected and 40% 1-RM conditions presented an increase in positive affect from baseline | Moderate |

(*Continued*)

**Table 1.** (Continued)

| Author(s) | Location | Design | Size | Mean age and standard-deviation | Intervention | Measures | Analysis | Outcomes | Bias risk |
|---|---|---|---|---|---|---|---|---|---|
| Greene & Petruzzello (2015) [55] | USA | Quasi-experimental study | $n = 22$ | $21.5 \pm 3.0$ | Training protocols at 70% and 100% 10-RM (randomly assigned) were completed on separate days. Both protocols included the same 7 exercises (bench press, leg curls, bent over rows, leg extensions, shoulders press, bicep curl, and triceps extension) performed for 3 sets of 10 repetitions 1-RM was measured in a previous session | FS measures were collected before, immediately after every set for every exercise, and at 5, 10, 15, and 20 minutes post training FAS and RPE were applied before, immediately after each exercise, and 20 minutes post training AD ACL; PACES; SA | MANOVA; repeated measures ANOVA; t test; Cohen's $d$ | The In-session positive affect in the 70% 10-RM protocol remained relatively high while in the 100% 10-RM protocol a decreased could be observed, only recovering after the end of the training session. Both protocols resulted in a post exercise increase in energy and calmness, while tiredness and anxiety decreased. The 70% 10-RM condition also resulted in a larger reduction in tension and higher reported enjoyment | Moderate |
| Hutchinson et al. (2020) [30] | USA | Quasi-experimental study | $n = 40$ | $35.0 \pm 9.2$ | Two sessions consisting in 3 sets of 10 repetitions of six exercises (leg press, hex bar deadlift, chest press, seated row, overhead press, and pulldown) in circuit, with an increase in intensity per circuit round (55%, 65% and 75% 1-RM) or the opposite (reverse order). 1-RM was measured in a previous session | FS assessment occurred during the last 10 s of both the work and recovery intervals (i.e., during the last 2–3 repetitions of the exercise and during the last 10 s of the 30-s recovery period) PRETIE-Q; PACES; EVS; RPE | Repeated-measures analysis of variance; Greenhouse-Geisser correction; repeated measures MANOVA | The increase in intensity condition resulted in a decrease in pleasure, while the decrease in intensity condition resulted in a slope of increasing pleasure and overall greater pleasure than the first condition The decrease in intensity condition also resulted in significantly greater post exercise pleasure, enjoyment of RT, and remembered pleasure | Moderate |
| Miller et al. (2009) [56] | USA | Quasi-experimental | $n = 31$ | $20.6 \pm 1.3$ | Three sessions with three different interventions: concentric, eccentric, and traditional RT. Three sets of chest press, seated row, overhead press, and biceps curl were performed at 80%, 100%, and 120% of 10-RM 10-RM was measured in a previous session | FS & FAS were measured before, immediately after, and 60 minutes after each exercise session AD ACL; RPE | Repeated-measures general linear model | All three interventions resulted in increases of core affect immediately after and 60 minutes after their conclusion | Moderate |

(*Continued*)

**Table 1.** (Continued)

| Author(s) | Location | Design | Size | Mean age and standard-deviation | Intervention | Measures | Analysis | Outcomes | Bias risk |
|---|---|---|---|---|---|---|---|---|---|
| Orssatto et al. (2020) [57] | Brazil | Quasi-experimental study | $n = 14$ | Males ($n = 7$): $27.1 \pm 6.0$ Females ($n = 7$): $28.3 \pm 5.7$ | Men and women were divided in two groups and performed two sessions of 6 sets, 12 repetitions, in a calf-raise machine to concentric failure 12-RM was measured in a previous session | FS & FAS were measured before exercise, and after each set PAAS; PACES; VAS; RPE-E; RPE-D | Repeated-measures ANOVA; Mauchly's test of sphericity; Greenhouse-Geisser correction; Bonferroni correction | Women reported displeasure and high activation after both exercise sessions, while men's affective response stayed in the low activation-pleasure quadrant | Moderate |
| Portugal et al. (2015) [58] | Brazil | Quasi-experimental study | $n = 16$ | $25.1 \pm 5.5$ | Four sessions performing 3 sets of 8 repetitions of 4 exercises (pulldown, leg extension, chest press and leg curl) at three prescribed intensities (40, 60, and 80% 1RM) and one self-selected intensity. 1-RM was measured in two previous sessions | FS & FAS were measured before exercise, immediately after the third set of each exercise, 10 minutes, and 20 minutes post session CR-10 | One-way ANOVA; repeated-measure ANOVA; Bonferroni correction | No significant differences in the affective response between the exercise groups; the 80% 1-RM group reached a negative affective response, but it returned to baseline at 10 and 20 minutes post; the leg curl exercise had the least positive affective response | Moderate |
| Richardson et al. (2018) [59] | England | Randomized Crossover Trial | $n = 10$ | Males ($n = 5$): $66 \pm 3$ Females ($n = 5$): $68 \pm 2$ | Participants completed three sets (chest press, leg press, calf raise, leg extension, leg curl, seated row, bicep curl, and triceps extension) of eight exercises on six separate occasions: three high-velocity, low-load (at 40% 1-RM) and three low-velocity, high load (at 80% 1-RM) sessions 1-RM was measured in a previous session | FS was measured prior to exercise and following every set of each exercise; FAS was measured before and after exercise RPE; PAAS; VAS; PACES | Repeated-measures ANOVA; Mauchly's test of sphericity; Huynh-Feldt adjustment; Greenhouse-Geisser correction; Bonferroni correction; *t*-test | FS & FAS ratings did not differ between conditions but did increase from pre- to post exercise | High |

(*Continued*)

**Table 1.** (Continued)

| Author(s) | Location | Design | Size | Mean age and standard-deviation | Intervention | Measures | Analysis | Outcomes | Bias risk |
|---|---|---|---|---|---|---|---|---|---|
| Richardson et al. (2020) [60] | England | Randomized Controlled Trial | $n = 40$ | HVLL1 ($n = 10$): 66 ± 5 LVHL ($n = 10$): 67 ± 4 HVLL2 ($n = 10$): 67 ± 6 LVHL2 ($n = 10$): 66 ± 6 | For 10 weeks, participants were assigned to either a high velocity, low load or low velocity, high load RT program either once or twice a week; each session consisted of three sets of eight exercises (leg press, seated row, chest press, leg extension, leg curl, calf raise, triceps extension, and bicep curl) performed for 14 repetitions at 40% 1-RM (both high velocity, low load conditions) or seven repetitions at 80% 1-RM (both low velocity, high load conditions) 1-RM was measured before the start of the intervention | FS & FAS were measured before and immediately after the exercise session in weeks 1, 5 and 10 RPE; PAAS; VAS; PACES | Repeated-measures ANOVA; one-way ANOVA; Mauchly's test of sphericity; Greenhouse-Geisser correction; Bonferroni correction; $t$-test | There were significant increases in affective response from before to after the exercise session but no differences between conditions | High |
| Schwartz et al. (2021) [61] | Israel | Quasi-experimental study | $n = 20$ | 34.4 ± 6.5 | Participants undertook two sessions of three sets of four exercises (leg-press, knee-extension, chest press, and lat pulldown) in either a predetermined condition (fixed for 10 repetitions in all sets) or by terminating the sets two repetitions from failure 1-RM was measured in a previous session | FS was applied before and after each set EES | Paired $t$-test; one sample $t$-test; mixed regression analysis | Both conditions elicited similar levels of affective valence, enjoyment, and approach preferences | Moderate |

(*Continued*)

**Table 1.** (Continued)

| Author(s) | Location | Design | Size | Mean age and standard-deviation | Intervention | Measures | Analysis | Outcomes | Bias risk |
|---|---|---|---|---|---|---|---|---|---|
| Stults-Kolehmainen et al. (2016) [62] | USA | Quasi-experimental study | $n = 57$ | 25.1 ± 5.5 | A two-phase, acute heavy-resistance exercise protocol: first phase consisting of a 10-RM leg press test and a second phase consisting of six sets at 80–100% of 10-RM. | FS & FAS were measured before the exercise sessions, at odd-numbered sets and after the last set of the first phase, and at sets 1, 3, 5, and 6 during the second phase | Pearson's product correlations; Kolmogorov-Smirnov test | Higher levels of stress were related to less affect | Serious |
| Tavares et al. (2020) [63] | Brazil | Randomized Crossover Trial | $n = 17$ | 24.5 ± 3.2 | 10RM protocol for bench press and knee extension to measure test re-test reliability in the first and second sessions In the third and fourth sessions subjects per-formed 4 sets of 10 repetitions of bench press and knee extension exercises with either a low tempo with 50% of 10RM or with a moderate tempo with 80% of 10RM | FS, FAS, RPE and attention focus after each set. | Repeated-measure two-way analysis of variance; Shapiro-Wilk test; Levene's test; Bonferroni correlation; Mauchly's test, Greenhouse-Geisser epsilon correction; Partial eta squared; paired t-test; Hedge's g | Even with a low load, the use of low tempo may not present advantages when the purpose is to enhance psychophysiological responses such as positive affective valence, lower activation, RPE, and attentional focus when compared with moderate tempo Affective valence decreased through the session while arousal displayed the opposite trend | Some concerns |

studies had an intervention/experimental design, including three randomized controlled trials, two randomized crossover trials, and 22 quasi-experimental studies. All studies used conve-nience methods of recruitment. Participants in the studies met the inclusion criteria set for this review, allowing for wider coverage of possible physical activity contexts and varying use of the FS and/or FAS in RT. In sum, the reviewed studies totaled a sample size (total N = 718); 17 studies were conducted with fewer than 30 participants (61%), eight studies had a sample between 30 and 50 individuals (31%), and two studies presented a sample size between 50 and 100 participants (8%).

## Risk of bias in studies

In the three randomized controlled trials included in this review, the overall risk of bias was of 'some concern' for Chang and Etnier [47] due to possible deviations from intended interven-tions and in the selection of the reported results; the other two trials [51, 60] were deemed at 'high risk' of bias due to the randomization process (see S2 File). As for the crossover trials a similar trend was observed, with Tavares et al. [63] being classified as of 'some concerns' due to possible deviations from intended interventions and in the selection of the reported results;

**Table 2. Summary of studies' and samples' characteristics.**

| Characteristics | Studies (%) | Samples K (%) |
|---|---|---|
| SAMPLE SIZE | 27 total | 718 total |
| <30 | 17 (63%) | 312 (44%) |
| 30–50 | 8 (30%) | 281 (39%) |
| 50–100 | 2 (7%) | 125 (17%) |
| SEX | | |
| Female only | 8 (30%) | 174 (24%) |
| Male only | 6 (22%) | 124 (17%) |
| Both sexes | 13 (48%) | 420 (59%) |
| LOCATION | | |
| North America | 10 (37%) | 356 (50%) |
| South America | 9 (33%) | 146 (20%) |
| Europe | 5 (19%) | 156 (22%) |
| Asia | 3 (11%) | 60 (8%) |
| MEAN AGE (years) | | |
| <18 | 1 (4%) | 11 (1%) |
| 18–64 | 22 (81%) | 637 (89%) |
| ≥65 | 4 (15%) | 70 (10%) |
| EFFECT SIZE AND/OR POWER CALCULATION | | |
| Yes | 13* (48%) | |
| No | 14 (52%) | |
| INSTRUMENTS APPLIED | | |
| FS only | 12 (44%) | |
| FAS only | 0 (0%) | |
| FS & FAS | 15 (56%) | |
| PRIOR TRAINING (FS/FAS) | | |
| Participants | 13 (48%) | |
| Researchers | 2 (8%) | |
| TIMING OF APPLICATION (FS and/or FAS) | Studies (%)** | Total measurements per RT Session |
| Before the session | 11 (41%) | 9 |
| During the set | 4 (15%) | 16 |
| Immediately after the set | 12 (44%) | 85 |
| After the set but unclear*** | 3 (11%) | 10 |
| Between sets | 14 (52%) | 126 |
| 5 min post exercise | 2 (7%) | 2 |
| 10 min post exercise | 3 (11%) | 3 |
| 15 min post exercise | 3 (11%) | 3 |
| 20 min post exercise | 2 (7%) | 2 |
| 30 min post exercise | 1 (4%) | 1 |
| 60 min post exercise | 1 (4%) | 1 |

Note. *One study failed to meet criteria due to dropouts (Portugal et al., 2015); **Percentage calculated considering the total number of studies included; ***Due to methodological issues (e.g., randomization of measures), the FS/FAS application timing could not be precisely defined

Richardson et al. [59] was deemed 'high risk' due to the randomization process (see S3 File). In the quasi-experimental studies, the overall bias was of 'moderate risk' for 15 studies [29, 30, 33, 45, 46, 48, 50, 52–58, 61] predominantly due to a possible bias in the measurement of outcomes, and selection of reported results. Seven studies were deemed 'serious risk' [26–28, 43,

44, 49, 62] also due to possible bias in the measurement of outcomes (see S4 File). Overall bias classification is presented in Table 1.

## Results of individual studies

**Contextual feasibility.**   Exercise intensity was established by using the one repetition maximum (1-RM) test in most of the studies (92%), and ranged from light (e.g., 40% 1-RM [54, 58]) to vigorous (e.g., 90% 1-RM [27]). Intensity prescription methods varied between self-selection by the participants [43, 44, 51, 54, 58], self-selection according to FS descriptors (e.g., adjusting the intensity to ensure the participants felt *'good'*–descriptor for '3' [50]), and by using a scale of repetitions in reserve [64] to measure the proximity to concentric failure [26]. Concentric task failure was reported in four studies [26, 29, 52, 57] for all exercises and in Chmelo et al. [48] for one exercise (crunches on a stability ball).

In terms of sampling 13 studies were comprised of males and females, eight studies included only females, and another six sampled only males (see Table 1). Orssatto et al. [57] is the only study depicting different affective responses between sex, with a group of females reporting displeasure when performing a calf-raise machine to muscle failure, conversely, males maintained a positive affective response. Most of the studies (81%) included adults (18–64 years), although in some studies the target population was the elderly [51, 53, 59, 60] or adolescents [43]. Regarding exercise experience, 12 studies sampled experienced individuals in RT activities, 10 sampled novice or inactive individuals, four did not report the sample's exercise experience [30, 43, 47, 63], and one choose to not control for exercise experience [56]. Lastly, 11 studies involved a combination of exercise machines and free weights for their RT interventions, while another 12 studies only made use of exercise machines, and four exclusively used free weights [27, 29, 47, 52]. Carraro et al. [33] reported that RT with free weights resulted in increased pleasantness and activation when compared with RT in machines, but these findings could not be replicated by Cavarretta et al. [28, 46].

**FS/FAS measurement procedures.**   A wide array of different timings of measurement combinations were used across all studies. No standardization in either the timing or the number of measurements could be disclosed. One study [26] noted the number of measurements as a potential issue; it was proposed that in a sample of recreational exercisers a single measurement can be enough to assess the affective response in an RT session prescribed to reach concentric failure. Most measurements were conducted during the RT session; specifically, immediately after the set (44%) and/or between sets (52%). It should be noted that when the measurement was not described as 'immediately after' a set it was indicated to have been recorded 'between sets'. Additionally, the timing of affect measurement could not be precisely defined in three studies [33, 45, 63]; predominantly, this was due to the methodological strategy of randomizing the order of multiple measures being collected at the same time point and on occasion, this resulted in other instruments being used before FS/FAS (e.g., perceived exertion, lactate threshold measurement). Four studies [27–30] measured the affective response during the set, while the muscles were under tension. Only one study did not directly measure the affective response during the RT session but applied the FS to adjust the exercise intensity according to predetermined descriptors [50]. Affective response was also (in addition to other assessment points during the session) measured outside of the RT session, either before [28, 45, 48, 50, 54–56, 58–60, 62], 5 min [33, 55], 10 min [45, 55, 58], 15 min [48, 54, 55], 20 min [55, 58], 30 min [28], and/or 60 min [56] after exercise.

Participants' training and/or familiarization prior to the use of the FS and/or FAS was reported in 13 studies. Of this number, prior training for the researchers that conducted the

data collection was only reported in two studies [26, 27]. Verbal encouragement was performed by the researchers in four studies [45, 57, 59, 62].

*Affective states plotted on the circumplex model of affect.* Of the 15 studies that used both the FS and the FAS, only six did not plot the affective response data in a circumplex model of affect [27, 47, 53, 55, 60, 62]. Generally, most of the studies [26, 45, 48, 58] that sampled individuals with RT experience showed a transition from the low-activation pleasure quadrant (i.e., calmness, relaxation) to the high-activation pleasure quadrant (i.e., energy, vigor). Orssatto et al. [57] is the only study that reported no transition to the high-activation pleasure quadrant with experienced exercisers; they reported a shift from low-activation pleasure to high-activation displeasure in a group of women, whilst men stayed in the low-activation pleasure quadrant despite a residual increase in activation. Two studies plotted the circumplex model with samples that were not considered to be experienced in RT. Specifically, Richardson et al. [59] reported a trend for higher pleasure and activation throughout the RT session, while Tavares et al. [63] reported the opposite trend with a shift from low-activation pleasure to high-activation displeasure.

## Discussion

The present study aimed to systematically analyze the use of the FS and/or the FAS in RT studies. The central focus was the contextual feasibility, timing, and frequency of measurement. In addition, the present study aimed to understand the implications for core affective response measurement in RT. A total of 27 studies met the inclusion criteria. The FS and/or FAS were used to assess the affective response in RT performed within a wide array of intensities (40% to 90% 1-RM), particularly in experienced exercisers, using both free weights and machines, male and female exercisers, and across a wide range of age (14–69 years). Overall, both scales appeared to be contextually appropriate for use in RT studies. However, several methodological issues were detected in the review of the studies regarding the timing and frequency of measurement, which may have implications for the interpretation of these results.

### Contextual feasibility, timing, and frequency of assessment

Considering the importance of exercise intensity in affect research [15], the requirement of an instrument that adequately assesses core affect in RT is of paramount importance. In this regard, the FS and the FAS appear to address this demand. Although vastly different exercise intensities could be observed across the 27 studies included in this review, the FS and FAS successfully measured affective responses regardless of the level of intensity or research/training protocol (i.e., 1-RM, self-selection, or repetitions in reserve). Additionally, the scales have been utilized in RT with varying proximities to concentric failure and were able to measure core affect even when an exercise was performed to the point of failure. Concerning sex representation, both sexes are well represented with the majority of the studies showing no clear differences in the affective response. Both scales were also applied not only to adults but also to adolescents and the elderly, demonstrating no age-related limitations in their interpretation and application. Lastly, no issue was reported when evaluating these instruments in RT sessions using free weights and/or machines, indicating core affect can be assessed independent of variations in exercise equipment.

One of the major concerns noted in the findings of the present review is the lack of standardization on the timing of measurement. In general, the timing of the measures was not based on a theoretical foundation, an evidence base, and/or recommendations presented in previous studies of affective responses. As a result, the variation in research practice impairs the ability to make comparisons across studies, extrapolate relatable findings, and develop

systematic guidelines. Current literature [15, 26, 27] suggests measures of affective responses should be obtained during or immediately after exercise, on the grounds that as time elapses between the behavior and the assessment moment, it becomes increasingly difficult to assess core affect without the interference of cognitive processes (i.e., possibly leading to an affective rebound). This suggests that measurements that are not applied during or immediately after exercise may fail to capture the true affective response to RT and present a distorted insight into the exerciser's experience. Four studies in the review demonstrated that measuring affect during exercise is feasible [27–30], and lend support to the affective assessment recommendations proposed by Stevens et al. [15]. Alternatively, 12 studies aligned with the recommendation to assess affective response immediately after a set [26, 27, 43, 46–48, 51, 54–56, 58, 59]. Although both approaches appear somewhat comparable in terms of feasibility, some considerations warrant discussion in the interpretation of the data derived for the alternative timings of assessment. For example, stopping mid-set (or at any point whilst the muscle is under tension) can affect the exercise dynamic (e.g., interrupting the intended exercise cadence and inducing an isometric contraction when only isotonic movement is wanted), influence the target objectives (e.g., increasing fatigue due to interruption in answering the scales), and have safety implications (e.g., diverting task attention from task execution under heavy load). Therefore, the use of scales immediately after the load removal/completion of a set can avoid mid-set disruption and record accurate data prior to a potential affective rebound effect [27].

Regarding the number of affective response measurements per RT session, the issue remains vastly unexplored. The number of data collection points in the studies reviewed were only aimed to be integrated into the study protocol, and they ranged from one to 60 assessments during the exercise session with no theoretical justification or attempts to provide recommendations for feasibility. Bastos et al. [26] is the only exception, highlighting that in apparently healthy and experienced exercisers a single core affect measurement may suffice to understand the pleasure/displeasure response. The matter of affective measurement frequency can be relevant in maximizing a pleasurable experience in RT. For example, in Portugal et al. [58], high-activation pleasure was the norm for the affective response in every condition except for the leg curl exercise when performed at 80% 1-RM, where core affect shifted to the high-activation displeasure quadrant. Failure to measure core affect frequently enough could result in such unpleasant responses being masked by the overall affective response of the RT session; conversely measuring too often could result in less ecological validity, less overall feasibility (e.g., reactivity to the test; variance carry-over effects), and burden the participant [32]. As such, the identification of recommendations in this regard that consider individual (e.g., exercise experience; health status) and exercise (e.g., intensity; volume) characteristics is paramount for the advancement of the affect regulation approach to exercise promotion.

## Other potentially relevant indicators

Considering the selected studies comprising this review, 15 studies used both the FS and the FAS, 12 studies used only the FS, and no study applied only the FAS. These results align with Stevens et al.'s [15] suggestion that much of the literature on affective responses to physical activity as a determinant of exercise adherence has predominantly focused on measuring core affective valence. According to the researchers that developed the FS [19] *'the scale was designed to evaluate the core emotions: pleasure / displeasure'* (p. 305). Thus, while the FAS measures levels of activation, the FS is necessary to differentiate positive activation from negative activation [15, 65]. Likewise, the FS alone cannot dissociate different states of positive (i.e., happy from calm) or negative (i.e., distressed from bored) affect from each other without the measurement of arousal [22]. This would suggest that some of the studies included in this

review may be limited in their interpretation of core affective responses by relying solely on one dimension of the affective response to RT.

Nine of the 15 studies that used both the FS and the FAS plotted the scores in a circumplex model to differentiate states of affective response. The application of the circumplex model offers a more comprehensive and fine-grained degree of affective states differentiation, acting as a map for the possible dynamic fluctuations across an RT exercise session [17, 22, 66], and allowing an idiographic and longitudinal approach to the affective response information.

When extracting and analyzing the data related to the included studies, some attention seems warranted in future research efforts on this topic. First, most of the literature presents small and convenient samples (i.e., N < 30 in 63% of the studies). Less than half of the studies (48%) conducted effect size and/or power calculations, risking confounding results due to a lack of statistical power. Second, only 13 studies (42%) reported participants received training in answering these scales, and only two (8%) reported these familiarization procedures for both researchers and participants. Moreover, verbal encouragement during data collection was reported in four studies, which can interfere with naturally occurring affective responses and subsequent comparisons across studies. This may suggest a heterogeneous approach to the training and preparation for these scales' use, and contrast with the suggestions made by Duda [67] and Evmenenko & Teixeira [17]. The knowledge of what is being assessed (i.e., feelings experienced *in the present moment*), as for the definition of subjective anchoring examples with the respondents, along with the ability of the researchers to provide them through contextual questioning, are important factors to achieve consistent application results. This methodological issue reflects a bias in data collection that must be addressed in future study efforts, particularly given the perceptual nature of core affect.

## Study limitations and future directions

This review is the first to explore the use of the FS/FAS to assess affective responses exclusively within RT activities; as such, it presents the most recent scientific evidence on how to use these instruments in studies undertaken with this mode of exercise. Nevertheless, this review is not without its limitations. First, the current literature on affective responses to RT is still scarce, this limits the extension of the extrapolation of the results. While the number of studies in this field of research is on the rise (of the 27 studies, 24 were published in the last 10 years, 15 in the last five years, and 10 in the last two years), more research grounded on current evidence of core affect measurements for this exercise type is warranted. Secondly, the heterogeneity of some sample characteristics as well as variability in methodological assessment approaches (e.g., timing), impairs the ability to interpret the recorded affective responses in RT. However, some consensus appears to be emerging in the most recently developed studies, and a suggestion on the timing of assessment (i.e., during or immediately after a set) is being used more consistently. Given the low number of studies that have focused on understanding how to promote quality measurement using both scales, replication studies will be paramount for this clarification and may help create a theoretically robust foundation for core affect assessment. Third, and still pertaining to the studies' heterogeneity, no clear indication of consistent practice emerged regarding the frequency of assessment. Although highly dependent on the sample characteristics (e.g., exercise experience), protocols to be tested (common RT session vs. circuit RT), and other exercise-related variables (e.g., session length), understanding which characteristics of RT will determine the frequency of assessment is paramount for the development of future affect regulation guidelines. Finally, although this review focuses on the FS and the FAS, it is important to remember that there are other instruments that may be of interest for affective response measurement. For example, the Activation Deactivation Adjective Check List

[68] represents another satisfactory option for affective response measurement in exercise, that also utilizes a circumplex model of affect [69].

## Conclusion

In sum, the FS and the FAS appear to be able to detect affective response changes within a wide array of load intensities, ages, equipment, and in both sexes. However, several methodological issues were observed. In particular, the timing and frequency of assessment should be considered in future research efforts when aiming to establish a more robust foundation for core affect assessment in RT. Given theoretical considerations, the limited evidence suggests that assessments during and immediately after a set are a valid approach for affect measurement, while the matter of the number of applications remains considerably unexplored.

Overall, the FS and FAS appear to be adequate scales for core affect assessment in RT, despite the heterogeneity in their application and some methodological issues detected. The combination of both scales may present advantages when aiming to understand the affective panorama during a RT session.

## Declarations

All authors contributed to the study's conception and design. The idea of this systematic review belongs to Diogo Santos Teixeira. Vasco Bastos performed the literature search, data analysis, and draft writing. All authors contributed to the risk of bias assessment. Diogo Santos Teixeira, Filipe Rodrigues, and Paul Davis made the final critical revision.

## Supporting information

**S1 Checklist. PRISMA 2020 statement compliance.**
(DOCX)

**S1 File. Sample MEDLINE search strategy.**
(DOCX)

**S2 File. Risk of bias assessment of the included randomized controlled trials.**
(TIF)

**S3 File. Risk of bias assessment of the included randomized crossover trials.**
(TIF)

**S4 File. Risk of bias assessment of the included quasi-experimental studies.**
(TIF)

## Acknowledgments

The authors would like to thank the researcher Leonor Henriques for the support given in the development of the study methodology.

## Author Contributions

**Conceptualization:** Filipe Rodrigues, Paul Davis, Diogo Santos Teixeira.

**Data curation:** Vasco Bastos, Filipe Rodrigues, Paul Davis, Diogo Santos Teixeira.

**Formal analysis:** Vasco Bastos, Filipe Rodrigues, Paul Davis, Diogo Santos Teixeira.

**Funding acquisition:** Filipe Rodrigues.

**Investigation:** Vasco Bastos, Filipe Rodrigues, Paul Davis, Diogo Santos Teixeira.

**Methodology:** Vasco Bastos, Filipe Rodrigues, Paul Davis, Diogo Santos Teixeira.

**Project administration:** Diogo Santos Teixeira.

**Resources:** Diogo Santos Teixeira.

**Supervision:** Diogo Santos Teixeira.

**Validation:** Filipe Rodrigues, Paul Davis, Diogo Santos Teixeira.

**Visualization:** Paul Davis, Diogo Santos Teixeira.

**Writing – original draft:** Vasco Bastos.

**Writing – review & editing:** Vasco Bastos, Filipe Rodrigues, Paul Davis, Diogo Santos Teixeira.

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
