## [Decision Letter · Decision Letter 0]

16 Aug 2023

PONE-D-23-08844Assessing affective valence and activation in resistance training with the feeling scale and the felt arousal scale: a systematic reviewPLOS ONE

Dear Dr. Davis,

Thank you for submitting your manuscript to PLOS ONE. After careful consideration, we feel that it has merit but does not fully meet PLOS ONE’s publication criteria as it currently stands. Therefore, we invite you to submit a revised version of the manuscript that addresses the points raised during the review process.

We look forward to receiving your revised manuscript.

Kind regards,

Preeti Kanawjia, MD

Academic Editor

PLOS ONE

Journal Requirements:

3. Please include a caption for figure 1.

Reviewers' comments:

Reviewer's Responses to Questions

**Comments to the Author**

1. Is the manuscript technically sound, and do the data support the conclusions?

Reviewer #1: Yes

Reviewer #2: Yes

Reviewer #3: Yes

2. Has the statistical analysis been performed appropriately and rigorously? 

Reviewer #1: Yes

Reviewer #2: N/A

Reviewer #3: N/A

3. Have the authors made all data underlying the findings in their manuscript fully available?

Reviewer #1: No

Reviewer #2: Yes

Reviewer #3: Yes

4. Is the manuscript presented in an intelligible fashion and written in standard English?

Reviewer #1: Yes

Reviewer #2: Yes

Reviewer #3: Yes

5. Review Comments to the Author

Reviewer #1: The submitted manuscript aimed to assess how the feeling scale and felt arousal scale have been used in resistance exercise, using a systematic review approach. In this effort, a pre-registration in PROSPERO was made, and the PRISMA recommendations were followed. Additionally, bias assessment instruments (e.g., ROBINS) were used to evaluate the quality of the studies that met the inclusion criteria. Generally, I commend the authors for the adequate methodological approach developed.

The manuscript focused on understanding of two well know scales used in the affective response studies in a relevant context of practice. Although a very specific topic of interest, this review may help advance the much-needed topic of how to use and assess affective responses.

Generally, I found the manuscript to be well-written and organized. The results suggest that the FA and FAS are useful but depict several concerns given the research produced on the topic. Although I have no major concerns about this manuscript, I would like the authors to clarify the following:

Introduction

How would sex be considered an individual characteristic to affect the FS and FAS use in RT? Shouldn’t these scales be interpreted the same way in both?

Method

Why an IMC > 34,9 was used? Does this contrast with the inclusion criteria of “apparently healthy individuals”’

Why effect size and power calculation were extracted from each study if no meta-analysis was performed?

There seems to be a typo in the ROBINS-I description (remove the dash in the extended form)

Discussion

“Although vastly different exercise intensities could be observed across the 27 studies included in this review, the FS and FAS successfully measured core affect”. In fact, this statement clashed with the following arguments that state that methodological issues impeded the assessment of core affect. Perhaps here you should say “the affective response” rather than “core affect”.

Apart from the FS and FAS, there are other instruments to assess the affective response. Should these be contemplated here (or in the limitations), as an alternative that can preset, perhaps, better support for its use?

Conclusion

Typo: the word “scales” in the first sentence is redundant. Also in this line, once again, perhaps saying “core affect” is not the accurate construct. Given that some studies applied the FS and FAS with other timings that may not be adequate, what the authors are mentioning here is the affective response.

Reviewer #2: The article touches on an interesting topic and is well written. As a suggestion:

1-Summary: I would change the discussion topic for results

2-Introduction: it is too long, it could be reduced in the main points of each subtopic and the objective, instead of using the name of the scale, could contain the outcome, which is affective responses

3-Methods:

3.1 In the selection process, how was the screening done? Was any specific application used to select titles, abstracts? It would be good to specify

3.2. It would be important to highlight the name of the application used for risk of bias, which is Robb

3.3 it would be interesting to put the risk of bias in a separate table

4- Results

4.1 it would be interesting to leave only the main points in the table and reduce the amount of text

4.2 It would be interesting to divide the tables by type of study, randomized and quasi-randomised. Also, I don't think it's necessary to put the type of study and statistical analysis. The table could also contain less information and be more objective

4.3 In the bias analysis results, it would be important to place the graphs that Robb generates, so that the general and individual biases of the included studies can be seen.

4.4 Was it not possible to perform a meta-analysis? It would be quite enriching. As the number of studies was not small, a more quantitative analysis of the data could be attempted.

4.5 it would be interesting to leave only the main points in results, because the details are already described in the tables

5. Discussion: could have commented more on the influence of exercise intensity on affective responses. The literature has shown that higher intensities have worse affective responses when compared to lower intensities.

Reviewer #3: This is a well-done study and a well-written manuscript. My only concern is as to why only two measured tools were included for affect in resistance training. A more robust approach including statistical analysis will help for better understanding as well as psychometric property assessment.

6. PLOS authors have the option to publish the peer review history of their article (what does this mean?). If published, this will include your full peer review and any attached files.

Reviewer #1: No

Reviewer #2: **Yes: **Ingrid Martins de França

Reviewer #3: No

---

## [Author Response · Author response to Decision Letter 0]

26 Aug 2023

Assessing affective valence and activation in resistance training with the feeling scale and the felt arousal scale: a systematic review

Response letter

Manuscript PONE-D-23-08844

Dear Editor-In-Chief of PLOS ONE,

Dr. Emily Chenette,

Thank you for the opportunity to resubmit our work and address the points raised during the review process.

Below you will find our response to the reviewers’ comments. As requested, we have also uploaded two versions of the revised manuscript, one with track changes and another without. 

We will be available for additional clarification whenever needed.

Best regards,

By all the authors,

The corresponding author.

Reviewer: 1

Comments to the Author

The manuscript focused on understanding of two well know scales used in the affective response studies in a relevant context of practice. Although a very specific topic of interest, this review may help advance the much-needed topic of how to use and assess affective responses.

Generally, I found the manuscript to be well-written and organized. The results suggest that the FA and FAS are useful but depict several concerns given the research produced on the topic. Although I have no major concerns about this manuscript, I would like the authors to clarify the following:

First, we would like to express our appreciation for the reviewer’s overall positive evaluation of our work and send our thanks for the valuable input. We are pleased to hear that the potential contribution of the review has been recognized and we have been successful in our articulation of the need for this study. We will endeavor to address all your concerns and will be available for further discussion on any of these topics.

Introduction

How would sex be considered an individual characteristic to affect the FS and FAS use in RT? Shouldn’t these scales be interpreted the same way in both?

Yes, we have not found in the literature indications suggesting that the FS and FAS would be interpreted differently between sexes. Additionally, the timing and frequency of the scales’ application have not been reported to require different approaches across individual characteristics. Possible differences in these scales’ scores are beyond the scope of this review but warrant future efforts for clarification.

Method

Why an IMC > 34,9 was used? Does this contrast with the inclusion criteria of “apparently healthy individuals”’

An individual with an BMI > 34.9 may not currently present any health conditions, as a high BMI is not an indicator of a health issue per se, but rather a status that may facilitate the development of issues. With this in mind, the inclusion criteria of “apparently healthy individuals” were crossed with other indications of health issues of the participants per study. When absent, the study sample was considered eligible. Additionally, considering these were a minority in the total sample of this review, no problems were expected to arise from adopting this approach.

Why effect size and power calculation were extracted from each study if no meta-analysis was performed?

This extraction was deemed necessary in order to present a state-of-the-art review regarding the sampling efforts of affective response research in RT. A total of 52% of the included studies did not conduct effect size or power calculation and ran the risk of undertaking underpowered analyses that can cause type II errors. This can result in considerable repercussions that may impede the advancement of this research field. It is our opinion that this limitation in the literature should be addressed, while simultaneously recommending future research runs these calculations in order to guarantee properly powered analyses.

There seems to be a typo in the ROBINS-I description (remove the dash in the extended form)

Thank you, the typo is now corrected.

Discussion

“Although vastly different exercise intensities could be observed across the 27 studies included in this review, the FS and FAS successfully measured core affect”. In fact, this statement clashed with the following arguments that state that methodological issues impeded the assessment of core affect. Perhaps here you should say “the affective response” rather than “core affect”.

Amendments have been made to the manuscript to correct this issue (page 26, lines 469-470):

“…the FS and FAS successfully measured affective responses regardless of the level of intensity or research/training protocol…”

Apart from the FS and FAS, there are other instruments to assess the affective response. Should these be contemplated here (or in the limitations), as an alternative that can preset, perhaps, better support for its use?

We agree that there are other adequate instruments to assess the affective response that can serve as alternatives. The present review focuses on the FS and FAS, although it can be warranted that other instruments may delineate the scope of this review. As such, we added to the manuscript the example of the Activation Deactivation Adjective Check List (pages 30-31, lines 607-613):

“Finally, although this review focuses on the FS and the FAS, it is important to remember that there are other instruments that may be of interest for affective response measurement. For example, the Activation Deactivation Adjective Check List [68] represents another satisfactory option for affective response measurement in exercise, that also utilizes a circumplex model of affect [69].”

Conclusion

Typo: the word “scales” in the first sentence is redundant. Also in this line, once again, perhaps saying “core affect” is not the accurate construct. Given that some studies applied the FS and FAS with other timings that may not be adequate, what the authors are mentioning here is the affective response.

Amendments have been made to correct these issues (page 31, line 609):

“In sum, the FS and the FAS appear to be able to detect affective response changes…”

Reviewer: 2

Comments to the Author

The article touches on an interesting topic and is well written.

We appreciate the positive evaluation of our manuscript and thank the reviewer for their valuable input. We will endeavor to address all your concerns and will be available for further discussion on any of these topics.

As a suggestion:

1-Summary: I would change the discussion topic for results

We agree that the abstract could have different wording for each topic/section and be equally viable. However, the Preferred Reporting Items for Systematic Reviews and Meta-Analyses (PRISMA) statement, which was followed for the development of this systematic review, states that the presented information should be found under a section titled “discussion” (please consult Table 2 of Page et al., 2020).

Page, M. J., McKenzie, J. E., Bossuyt, P. M., Boutron, I., Hoffmann, T. C., Mulrow, C. D., ... & Moher, D. (2021). The PRISMA 2020 statement: an updated guideline for reporting systematic reviews. International journal of surgery, 88, 105906.

2-Introduction: it is too long, it could be reduced in the main points of each subtopic and the objective, instead of using the name of the scale, could contain the outcome, which is affective responses

We agree that the Introduction is extensive; the length of the Introduction was required to comprehensively explore the rationale and objectives of this systematic review. Considering the complexity of the subject at hand, a less thorough Introduction could fail to elucidate the constructs and enlighten the reader about the importance of, for example, the timing of affective response assessment.

Regarding the usage of the name of the scales, it was deemed necessary in order to avoid confusion with other instruments that also measure affective response. We do acknowledge that this approach results in a longer Introduction, yet this is necessary to facilitate comprehension and appreciation of the scope of the review.

3-Methods:

3.1 In the selection process, how was the screening done? Was any specific application used to select titles, abstracts? It would be good to specify

No application or software was utilized and the screening process was conducted manually. Amendments have been made in the manuscript to clarify this process (page 8, lines 230-231):

“…identified records from the database search were manually screened, analyzed, and checked against eligibility criteria.”

Of note, although machine learning might be a new and exciting search method for systematic reviews, there is currently no evidence demonstrating its superiority over traditional manual search strategies (Dos Reis et al., 2023; Orgeolet et al., 2020). 

Dos Reis, A. H. S., de Oliveira, A. L. M., Fritsch, C., Zouch, J., Ferreira, P., & Polese, J. C. (2023). Usefulness of machine learning softwares to screen titles of systematic reviews: a methodological study. Systematic Reviews, 12(1), 1-14.

Orgeolet, L., Foulquier, N., Misery, L., Redou, P., Pers, J. O., Devauchelle-Pensec, V., & Saraux, A. (2020). Can artificial intelligence replace manual search for systematic literature? Review on cutaneous manifestations in primary Sjögren’s syndrome. Rheumatology, 59(4), 811-819.

3.2. It would be important to highlight the name of the application used for risk of bias, which is Robb

We agree and have made amendments to the manuscript accordingly (page 9, lines 271-273):

“Lastly, the robvis online application [37] was used to create figures that present a more detailed overview of each included study’s risk of bias (S2-S4 Files).”

3.3 it would be interesting to put the risk of bias in a separate table

Thank you for the suggestion. We address this concern in your comment 4.3.

4- Results

4.1 it would be interesting to leave only the main points in the table and reduce the amount of text

When taking into account the objective of the present systematic review and the limitations in the literature it addresses, we believe that the information presented in Table 1 is necessary for the proper interpretation of the results and discussion sections that follow. For example, properly describing each study’s intervention is necessary to truly comprehend how and when the FS and the FAS were utilized. These descriptions are already brief, further reductions would risk leaving out relevant details that are important for interpretation. Additionally, other systematic reviews in the literature present tables of similar size and information display (e.g., Cavarretta et al., 2018; Ekkekakis et al., 2011; Evmenenko & Teixeira, 2020; Henriques & Teixeira, 2023).

Cavarretta, D. J., Hall, E. E., & Bixby, W. R. (2018). The acute effects of resistance exercise on affect, anxiety, and mood–practical implications for designing resistance training programs. International Review of Sport and Exercise Psychology, 12(1), 295-324.

Ekkekakis, P., Parfitt, G., & Petruzzello, S. J. (2011). The pleasure and displeasure people feel when they exercise at different intensities: decennial update and progress towards a tripartite rationale for exercise intensity prescription. Sports medicine, 41, 641-671.

Evmenenko, A., & Teixeira, D. S. (2022). The circumplex model of affect in physical activity contexts: a systematic review. International Journal of Sport and Exercise Psychology, 20(1), 168-201.

Henriques, L., & Teixeira, D. S. (2023). Assessing Affective Valence and Activation in Stretching Activities with the Feeling Scale and the Felt Arousal Scale: A Systematic Review. Perceptual and Motor Skills, 130(3), 1099-1122.

4.2 It would be interesting to divide the tables by type of study, randomized and quasi-randomised. Also, I don't think it's necessary to put the type of study and statistical analysis. The table could also contain less information and be more objective

We agree that a division by study type is a viable method to organize a table. However, the present review did not particularly emphasize the types of studies presented in the literature. Additionally, most of the included studies were quasi-experimental (with only three randomized controlled trials and two randomized crossover trials). As such, organizing Table 1 following this criterion could be redundant and ultimately confusing for the reader. Other systematic reviews also present a similar table organization by author’s surname (e.g., Cavarretta et al., 2018; Ekkekakis et al., 2011; Evmenenko & Teixeira, 2020; Henriques & Teixeira, 2023).

Concerning the presence of the type of study and statistical analysis information in Table 1, while we agree that it is not indispensable, we believe it provides further elucidation for the reader regarding each study’s methodological approach. 

Cavarretta, D. J., Hall, E. E., & Bixby, W. R. (2018). The acute effects of resistance exercise on affect, anxiety, and mood–practical implications for designing resistance training programs. International Review of Sport and Exercise Psychology, 12(1), 295-324.

Ekkekakis, P., Parfitt, G., & Petruzzello, S. J. (2011). The pleasure and displeasure people feel when they exercise at different intensities: decennial update and progress towards a tripartite rationale for exercise intensity prescription. Sports medicine, 41, 641-671.

Evmenenko, A., & Teixeira, D. S. (2022). The circumplex model of affect in physical activity contexts: a systematic review. International Journal of Sport and Exercise Psychology, 20(1), 168-201.

Henriques, L., & Teixeira, D. S. (2023). Assessing Affective Valence and Activation in Stretching Activities with the Feeling Scale and the Felt Arousal Scale: A Systematic Review. Perceptual and Motor Skills, 130(3), 1099-1122.

4.3 In the bias analysis results, it would be important to place the graphs that Robb generates, so that the general and individual biases of the included studies can be seen.

Thank you for the suggestion. We agree and have included the figures as Supplementary Material. 

4.4 Was it not possible to perform a meta-analysis? It would be quite enriching. As the number of studies was not small, a more quantitative analysis of the data could be attempted.

We agree that statistical procedures such as meta-analysis are very interesting to reinforce a systematic review with quantitative data. However, given the considerable heterogeneity in the methodological approach across the studies, a meta-analysis could present some issues (Lee, 2019; Walker et al., 2008). With such variability in the timing of affective response measurement, doubts can arise if what was truly measured was, in fact, core affect in RT, or something else (e.g., how one feels now that the exercise is over; a recall of the exercise that involves cognitive processes). At this time, we fear that such a statistical procedure would add to the problem. Instead, we focused on providing core affect measurement recommendations for future research, hoping that, in the future, a meta-analysis without significant heterogeneity problems can be conducted. 

Lee, Y. H. (2019). Strengths and limitations of meta-analysis. The Korean Journal of Medicine, 94(5), 391-395.

Walker, E., Hernandez, A. V., & Kattan, M. W. (2008). Meta-analysis: Its strengths and limitations. Cleveland Clinic journal of medicine, 75(6), 431.

4.5 it would be interesting to leave only the main points in results, because the details are already described in the tables

The points presented in the Results section intend to further elucidate and explore the information presented in Tables 1 and 2 according to the review’s objectives. Additionally, the information that is reported follows the current PRISMA statement (please see Table 1 in Page et al., 2020).

Page, M. J., McKenzie, J. E., Bossuyt, P. M., Boutron, I., Hoffmann, T. C., Mulrow, C. D., ... & Moher, D. (2021). The PRISMA 2020 statement: an updated guideline for reporting systematic reviews. International journal of surgery, 88, 105906.

5. Discussion: could have commented more on the influence of exercise intensity on affective responses. The literature has shown that higher intensities have worse affective responses when compared to lower intensities.

The close relationship between affective response and exercise intensity is in fact paramount for exercise adherence. This relationship is outlined in the Introduction section when addressing the importance of the timing of assessment due to a possible rebound effect (lines 108-112): “In particular, the timing of assessment is of notable concern due to a possible ‘affective rebound’ phenomenon (i.e., improvement of the affective response after exercise termination) that has been well documented in aerobic activities [17], as well as emerging evidence from the limited studies on RT [29,30,31]”. The importance of exercise intensity is once again mentioned in the Discussion section (lines 351 – 353): “Considering the importance of exercise intensity in affect research [15], the requirement of an instrument that adequately assesses core affect in RT is of paramount importance”. As noted in the present review, this affect-intensity relationship and its implications were always addressed when deemed necessary for its objectives. Given the complexity of this relationship (e.g., interindividual variance; Ekkekakis et al., 2005), we believe that further exploring this phenomenon would be outside of the scope of this review. Additionally, even if it was not, the heterogeneity found in the timing of assessment would render this task rather problematic.

Ekkekakis, P., Hall, E. E., & Petruzzello, S. J. (2005). Some like it vigorous: Measuring individual differences in the preference for and tolerance of exercise intensity. Journal of Sport and Exercise Psychology, 27(3), 350-374.

Reviewer: 3 

Comments to the Author

This is a well-done study and a well-written manuscript. 

We appreciate the positive evaluation of our manuscript and thank the reviewer for the valuable input. We will endeavor to address your concerns and will be available for further discussion on any of these topics.

My only concern is as to why only two measured tools were included for affect in resistance training. 

The FS and the FAS are valid and reliable scales to measure the core affect dimensions of affective valence (Hardy & Rejesky, 1989) and arousal (Svebak & Murgatroyd, 1985). Both scales have been extensively used in the literature to measure affective response in exercise due to being relatively quick and effortless to use (Evmenenko & Teixeira, 2020), and have been suggested or recommended to be used for this purpose (ACSM, 2021; Ekkekakis 2013; Stevens et al., 2020). Additionally, a similar reasoning was used recently to explore the affective response with the FS and FAS in stretching activities (Henriques & Teixeira, 2023), highlighting that although other instruments could be used for this purpose, the FS and FAS are indeed useful and more widely recognized for this endeavor, and may in the future received support for its use in a wide array of activities. 

Additionally, another systematic review by Cavarretta et al. (2018) focused on the effects of RT on affect, anxiety, and mood. This review offered a broad exploration of affect in RT activities, including a wide array of instruments used for this endeavor. We believe that not enough time has passed since this publication to warrant another broader review on the subject without risking redundancy. Instead, this review focuses on the instruments used to measure one of the most relevant affect variables for exercise adherence (i.e., core affect; Stevens et al., 2020). 

American College of Sports Medicine. (2021). ACSM's guidelines for exercise testing and prescription. Lippincott Williams & Wilkins.

Cavarretta, D. J., Hall, E. E., & Bixby, W. R. (2018). The acute effects of resistance exercise on affect, anxiety, and mood–practical implications for designing resistance training programs. International Review of Sport and Exercise Psychology, 12(1), 295-324.

Ekkekakis, P. (2013). The measurement of affect, mood, and emotion: A guide for health-behavioral research. Cambridge University Press.

Evmenenko, A., & Teixeira, D. S. (2022). The circumplex model of affect in physical activity contexts: a systematic review. International Journal of Sport and Exercise Psychology, 20(1), 168-201.

Hardy, C. J., & Rejeski, W. J. (1989). Not what, but how one feels: the measurement of affect during exercise. Journal of sport and exercise psychology, 11(3), 304-317.

Stevens, C. J., Baldwin, A. S., Bryan, A. D., Conner, M., Rhodes, R. E., & Williams, D. M. (2020). Affective determinants of physical activity: a conceptual framework and narrative review. Frontiers in Psychology, 3366.

Henriques, L., & Teixeira, D. S. (2023). Assessing Affective Valence and Activation in Stretching Activities with the Feeling Scale and the Felt Arousal Scale: A Systematic Review. Perceptual and Motor Skills, 130(3), 1099-1122.

Svebak, S., & Murgatroyd, S. (1985). Metamotivational dominance: a multimethod validation of reversal theory constructs. Journal of personality and social psychology, 48(1), 107.

A more robust approach including statistical analysis will help for better understanding as well as psychometric property assessment.

We agree that statistical procedures such as meta-analysis are very interesting to reinforce a systematic review with quantitative data. However, given the considerable heterogeneity in the methodological approach across the studies, a meta-analysis could present some issues (Lee, 2019; Walker et al., 2008). With such variability in the timing of affective response measurement, doubts can arise if what was truly measured was, in fact, core affect in RT, or something else (e.g., how one feels now that the exercise is over; a recall of the exercise that involves cognitive processes). We fear that such a statistical procedure would add to the problem that is noted across the studies comprising the review. Instead, we focused on providing core affect measurement recommendations for future research, hoping that, in the future, a meta-analysis without significant heterogeneity problems can be conducted.

Regarding psychometric property assessment, recent research in our group has focused on the validity and reliability of the FS and FAS (Brito et al., 2022).

Brito, H., Teixeira, D., & Araújo, D. (2022). Translation and Construct Validity of the Feeling Scale and the Felt Arousal Scale in Portuguese Recreational Exercisers. Cuadernos de Psicología Del Deporte, 22(3), 103-113.

Lee, Y. H. (2019). Strengths and limitations of meta-analysis. The Korean Journal of Medicine, 94(5), 391-395.

Walker, E., Hernandez, A. V., & Kattan, M. W. (2008). Meta-analysis: Its strengths and limitations. Cleveland Clinic journal of medicine, 75(6), 431.

---

## [Decision Letter · Decision Letter 1]

3 Nov 2023

Assessing affective valence and activation in resistance training with the feeling scale and the felt arousal scale: a systematic review

PONE-D-23-08844R1

Dear Dr. Davis,

We’re pleased to inform you that your manuscript has been judged scientifically suitable for publication and will be formally accepted for publication once it meets all outstanding technical requirements.

Kind regards,

Preeti Kanawjia, MD

Academic Editor

PLOS ONE

Additional Editor Comments (optional):

Reviewers' comments:

Reviewer's Responses to Questions

**Comments to the Author**

1. If the authors have adequately addressed your comments raised in a previous round of review and you feel that this manuscript is now acceptable for publication, you may indicate that here to bypass the “Comments to the Author” section, enter your conflict of interest statement in the “Confidential to Editor” section, and submit your "Accept" recommendation.

Reviewer #1: All comments have been addressed

2. Is the manuscript technically sound, and do the data support the conclusions?

Reviewer #1: Yes

3. Has the statistical analysis been performed appropriately and rigorously? 

Reviewer #1: N/A

4. Have the authors made all data underlying the findings in their manuscript fully available?

Reviewer #1: Yes

5. Is the manuscript presented in an intelligible fashion and written in standard English?

Reviewer #1: Yes

6. Review Comments to the Author

Reviewer #1: I would Like to congrats the authors for addressing all of my previous comments. Therefore, the paper met the conditions to be accepted.

7. PLOS authors have the option to publish the peer review history of their article (what does this mean?). If published, this will include your full peer review and any attached files.

Reviewer #1: No

---

## [Editor Report · Acceptance letter]

9 Nov 2023

PONE-D-23-08844R1 

Assessing affective valence and activation in resistance training with the feeling scale and the felt arousal scale: a systematic review 

Dear Dr. Davis:

I'm pleased to inform you that your manuscript has been deemed suitable for publication in PLOS ONE. Congratulations! Your manuscript is now with our production department. 

Kind regards, 

on behalf of

Dr. Preeti Kanawjia 

Academic Editor

PLOS ONE